# Association of Pre- and Gestational Conditions and Barriers to Breastfeeding with Exclusive Breastfeeding Practices

**DOI:** 10.3390/nu17142309

**Published:** 2025-07-13

**Authors:** Reyna Sámano, Gabriela Chico-Barba, Hugo Martínez-Rojano, María Eugenia Mendoza-Flores, María Hernández-Trejo, Carmen Hernández-Chávez, Andrea Luna-Hidalgo, Estefania Aguirre-Minutti, Ricardo Gamboa, María Eugenia Flores-Quijano, Otilia Perichart-Perera, Andrea López-Ocampo

**Affiliations:** 1Coordinación de Nutrición y Bioprogramación, Instituto Nacional de Perinatología, Secretaría de Salud Montes Urales 800, Lomas de Virreyes, Alcaldía Miguel Hidalgo, Mexico City 11000, Mexico; gabyc3@gmail.com (G.C.-B.); tina14mx@yahoo.com (M.E.M.-F.); maria.h.trejo72@gmail.com (M.H.-T.); alunahgo@gmail.com (A.L.-H.); minuttiestefania@gmail.com (E.A.-M.); maru_fq@yahoo.com (M.E.F.-Q.); oti_perichart@yahoo.com (O.P.-P.); 2Sección de Posgrado e Investigación de la Escuela Superior de Medicina del Instituto Politécnico Nacional, Plan de San Luis y Díaz Mirón s/n, Colonia Casco de Santo Tomas, Alcaldía Miguel Hidalgo, Mexico City 11340, Mexico; 3Programa de Maestría en Ciencias de la Salud, Escuela Superior de Medicina del Instituto Politécnico Nacional, Plan de San Luis y Díaz Mirón s/n, Colonia Casco de Santo Tomas, Alcaldía Miguel Hidalgo, Mexico City 11340, Mexico; 4Departamento de Neurobiología del Desarrollo, Instituto Nacional de Perinatología, Secretaría de Salud Montes Urales 800, Lomas de Virreyes, Alcaldía Miguel Hidalgo, Mexico City 11000, Mexico; karmenhdez@hotmail.com; 5Escuela de Dietética y Nutrición del ISSSTE, Callejón Vía, Av. San Fernando No. 12, San Pedro Apóstol, Tlalpan, Mexico City 14070, Mexico; 6Departamento de Fisiología, Instituto Nacional de Cardiología “Ignacio Chávez”, Mexico City 14080, Mexico; rgamboaa_2000@yahoo.com; 7Facultad de Nutrición, Universidad Autónoma del Estado de Morelos, Cuernavaca 62350, Mexico; andrealooc1010@gmail.com

**Keywords:** breastfeeding, barriers, facilitators, high-risk pregnancies, exclusive breastfeeding, non-communicable diseases, food insecurity

## Abstract

Background: Breastfeeding is essential for preventing non-communicable diseases. However, mothers with chronic illnesses tend to breastfeed less, increasing the likelihood of abandoning breastfeeding, especially if they experience gestational complications. Objective: To analyze the association between factors such as prepregnancy maternal characteristics, gestational complications, food security, barriers, and facilitators with the practice of exclusive breastfeeding. Methods: Cross-sectional study with 566 women who had prenatal care and gave birth at the National Institute of Perinatology (Mexico City) between 2021 and 2024. Surveys were administered on breastfeeding practices, food insecurity, barriers, and facilitators of exclusive breastfeeding in mothers. In addition, sociodemographic information, medical history (prepregnancy conditions and complications), gestational weight gain, and neonatal outcomes were recorded. Results: Of the 566 women, only 43.6% practiced exclusive breastfeeding, with a median duration of 4 months. Exclusive breastfeeding was more frequent in young, stay-at-home mothers with lower educational attainment and in those with food insecurity, who also tended to delay the introduction of complementary foods until after six months. Prepregnancy risk conditions (OR 1.56, 95% CI 1.06–2.30) and multiparity (OR 1.64, 95% CI 1.08–2.49) increased the risk of non-exclusive breastfeeding. Conversely, food insecurity (OR 0.40, 95% CI 0.20–0.78) and counseling from healthcare personnel (OR 0.09, 95% CI 0.01–0.51) showed a protective effect. The analysis also showed that paid employment (OR 4.68, 95% CI 1.65–13.21), the perception of low milk production (OR 6.45, 95% CI 2.95–14.10), maternal illness/medication (OR 3.91, 95% CI 1.36–11.28), and fatigue (OR 4.67, 95% CI 1.36–11.28) increased the probability of non-exclusive breastfeeding. Conclusions: In Mexico, the promotion of exclusive breastfeeding faces challenges, especially in mothers who begin pregnancy with significant chronic conditions such as diabetes, hypertension, obesity, advanced maternal age, and hypothyroidism, among others. Healthcare personnel should provide personalized advice to each woman from the prenatal stage on strategies to achieve and maintain exclusive breastfeeding.

## 1. Introduction

Breastfeeding offers significant benefits for both mother and baby, thanks to the essential nutrients present in breast milk [1]. Exclusive breastfeeding is associated with a range of positive outcomes in infant health. In the short term, it reduces the length of hospital stays and promotes healthy weight gain. In the long term, it contributes to healthier eating habits, a lower body mass index and adiposity, lower cholesterol levels, and improved cognitive and behavioral development. Even in children with metabolic disorders, exclusive breastfeeding helps maintain metabolic stability [2]. Breastfeeding with breast milk offers significant advantages for mothers, such as a lower risk of developing certain types of cancer and better metabolic and cardiovascular health [3]. Babies fed with breast milk have a lower risk of sudden infant death syndrome and developing type 2 diabetes later in life. There is also evidence that breastfeeding can help prevent overweight and obesity in preschool children [4].

Evidence indicates that the decline in breastfeeding practices is related to an increase in chronic diseases such as diabetes, heart problems, obesity, and autoimmune diseases [5]. Previous research has shown that breastfeeding is linked to a lower risk of type 2 diabetes, both for healthy mothers and those who had diabetes during pregnancy [6]. A study by Suthasmalee and Phaloprakarn observed that the longer a mother breastfeeds her child, the lower her risk of developing diabetes after delivery [7].

Although breastfeeding is a natural process, it requires practice and learning. Some women face difficulties initiating and maintaining lactation and therefore need support from a healthcare professional specialized in breastfeeding [8]. For example, women with diabetes may experience greater challenges due to a delay in the onset of copious milk production after delivery [9].

Women who experience high-risk pregnancies are less likely to exclusively breastfeed their babies, and the duration of breastfeeding may be shorter compared to those with normal pregnancies [10,11,12]. Several barriers have been identified that could hinder successful breastfeeding in the case of high-risk pregnancies, such as an increase in cesarean sections, premature births, premature rupture of membranes, separation between mother and child at birth, and the late initiation of breastfeeding, in addition to the early introduction of formula [13,14].

The initiation of breastfeeding can be particularly difficult for women with pre-existing comorbidities or who have experienced complications during pregnancy [15]. Therefore, it is crucial that they receive consistent and evidence-based information to support early feeding practices adapted to their needs [16]. The use of commercial formula milk should be avoided unless there is a precise medical indication [17]. Furthermore, it is essential to offer specific solutions for lactation problems related to pre-existing diseases or gestational complications. Women with high-risk pregnancies who choose to breastfeed require individualized lactation plans to maximize their chances of success [18]. This will promote the best clinical practices within multidisciplinary healthcare teams.

While breastfeeding is essential for the prevention and control of non-communicable diseases, women with certain health conditions may encounter greater obstacles to initiating and maintaining it. Research on the relationship between high-risk pregnancies and breastfeeding outcomes is limited, which hinders our understanding of this connection. This study seeks to address this gap by comparing breastfeeding outcomes between mothers with high-risk pregnancies and healthy mothers. In addition, it aims to analyze the association between factors such as maternal prepregnancy characteristics, gestational complications, food security, barriers, and facilitators with the practice of exclusive breastfeeding.

## 2. Methods

### 2.1. Study Design and Setting

A cross-sectional study was conducted with a sample of 566 women. All participants completed the surveys, received prenatal care, and delivered their babies at the Instituto Nacional de Perinatología (INPer), which is a tertiary referral institution located in Mexico City specializing in the care of women with high-risk pregnancies, who constitute the majority of its patient population. Data collection took place between January 2021 and December 2024.

### 2.2. Participants

The study population was selected from women followed from the third trimester of pregnancy to six months postpartum. The source population consisted of women receiving prenatal care at the National Institute of Perinatology in Mexico City. Non-probability sampling of consecutive cases meeting the selection criteria was used until the calculated sample size was reached.

Women were classified according to (a) those who had one or more prepregnancy conditions that posed a real or potential risk to maternal or fetal well-being but did not develop any other complications during pregnancy; (b) those with no history of prepregnancy conditions who experienced one or more complications during pregnancy, childbirth, or the postpartum period; (c) those with one or more prepregnancy conditions who also experienced a complication during pregnancy, childbirth, or the postpartum period; and (d) those who had no prepregnancy conditions and who also did not experience any complications during pregnancy, childbirth, or the postpartum period (see Figure 1).

Participants were included in the study if they had a high-risk pregnancy, defined as any condition posing an actual or potential risk to the health of the mother or fetus. Specific conditions for inclusion encompassed a wide range of factors, including adolescent pregnancy, various forms of diabetes and heart disease, polycystic ovary syndrome, infertility, obesity, addiction, kidney and autoimmune diseases, lung and liver conditions, HIV, neurological disorders, severe anemia (hemoglobin < 7.5 g/dL) requiring hospitalization and blood transfusion, thyroid issues, multiple pregnancies, and various pregnancy-related complications.

The exclusion criteria were pregnant women with less than 27 weeks of gestation and outside the reproductive age (defined as under 15 or over 45 years). Participants were required to have a live newborn and agree to attend at least two appointments for data collection. Confirmation of gestational age and reproductive age was obtained from medical records.

Each eligible participant received an individual tracking card with the complete schedule, indicating the dates and times of the appointments. They received a phone call to remind them of their appointment or to address any questions they might have. During these sessions, their breastfeeding habits and the obstacles they encountered while breastfeeding were analyzed.

In the first appointment, conducted at the end of the third trimester, sociodemographic information (telephone numbers and address) was obtained to facilitate contact for the second appointment. Six months after delivery, each participant was contacted to schedule the second appointment. Questionnaires designed to assess barriers and facilitators related to breastfeeding practice and on food insecurity were administered at this appointment. To ensure the information was obtained, research staff contacted the participant by telephone. If the participant was unable to attend the second appointment, a video call was made to obtain the essential information. If the mother was unable to make the video call, research staff from the study visited the mother’s home to conduct the survey.

### 2.3. Data Collection

For data collection, a structured questionnaire was designed based on a comprehensive review of the relevant scientific literature. The questionnaire contained multiple-choice and selection-based questions about barriers and facilitators to breastfeeding. These questions aimed to identify the specific factors that influenced the duration of breastfeeding.

Before its formal implementation, the questionnaire underwent a pilot test with 30 women whose demographic and clinical characteristics were similar to those of the study’s target population. Participants in the pilot test were excluded from the final analysis. The purpose of this pilot test was to evaluate the clarity, comprehension, and acceptability of the questions, as well as to estimate the average time required for its administration.

Based on the results of the pilot test, adjustments and improvements were made to the questionnaire to optimize its content validity and feasibility. The reliability of the questionnaire was evaluated using Cronbach’s alpha coefficient, yielding a value of 0.830, which indicates adequate internal consistency. Additionally, content validity was evaluated using the agreement index (agreements out of the total responses), with a value of 0.87.

Questions designed to assess knowledge about breastfeeding were presented in multiple-choice format. On the other hand, questions about breastfeeding practices were closed-ended (dichotomous or Likert scale), with the possibility for participants to provide additional explanations or comments when they deemed necessary.

Additionally, another questionnaire was applied that included questions designed to collect information on sociodemographic characteristics of the participants, events and complications related to childbirth, knowledge and practices of breastfeeding, sources of information about breastfeeding, and perceived family support in relation to breastfeeding, together with a section on food insecurity.

### 2.4. Sociodemographic Information

Sociodemographic data were collected during the third trimester of pregnancy and recorded on a standardized form. Age was obtained through direct questioning and categorized according to obstetric risk into three groups: ≤19 years, 20 to 34.9 years, and ≥35 years. Education level was recorded in years and classified as completed secondary school, high school, and university studies, according to the Mexican educational system. Marital status was classified into two categories: married/common-law union and single (including widowed, divorced, and separated). Occupation was categorized as homemakers and paid workers (those who obtained economic remuneration for their work outside the home). Socioeconomic status was determined by applying an 8-question survey based on the methodology of the Mexican Association of Market Intelligence Agencies (AMAI), which assesses aspects related to housing, education, and family assets. The results were grouped into six levels, from highest to very low [19].

### 2.5. Pre- and Gestational Conditions

A high-risk pregnancy was defined as one in which certain factors increase maternal and/or fetal morbidity and mortality, with an increased likelihood of complications or unfavorable outcomes for both mother and baby. These factors may be biological, socioeconomic, or environmental. In this study, a woman was considered to have a high-risk pregnancy if her medical history indicated prepregnancy conditions that posed a risk to the health of both mother and fetus [20]. These conditions included, but were not limited to, advanced maternal age; teenage pregnancy; type 1 or type 2 diabetes mellitus, or gestational diabetes mellitus; heart disease (rheumatic or valvular heart disease, essential hypertension, or pregnancy with concomitant hypertension or preeclampsia); polycystic ovary syndrome; infertility; morbid obesity; drug addiction; kidney disease; autoimmune diseases; coagulopathies; chronic lung disease (bronchial asthma or chronic tuberculosis); chronic hepatitis (viral hepatitis); hepatitis C virus infection; human immunodeficiency virus (HIV); neurological disease (epilepsy, among others); anemia (hemoglobin < 7.5 g/dL) requiring hospitalization and blood transfusion; thyroid disease (hypothyroidism or hyperthyroidism); multiple pregnancies; cervical insufficiency; placenta previa; recurrent urinary and kidney infections; premature rupture of membranes; preterm delivery; macrosomia; low birth weight; and other conditions that could increase maternal or fetal mortality. Information on prepregnancy and pregnancy-related conditions (complications) was obtained from participants’ medical records and corroborated with laboratory test results [21].

### 2.6. Gestational Weight Gain

Weight and height measurements, used to calculate gestational weight gain, were obtained one or two weeks before delivery. In cases of premature delivery, this information was extracted from the nursing note at the time of the woman’s admission in labor. Prepregnancy weight was obtained through direct questioning of the participant; the literature has demonstrated the reliability of self-reported prepregnancy weight. Gestational weight gain was calculated using the methodology proposed by Adu-Afarwuah S et al. [22] in which prepregnancy BMI was used to calculate the percentage of adequacy according to gestational age. Weight gain was then classified as insufficient, adequate, or excessive. In the case of adolescents, prepregnancy BMI was calculated and classified according to the World Health Organization (WHO) 2006 guidelines [23].

### 2.7. Assessment of Breastfeeding Practice

To assess breastfeeding practice, the number of months in which the baby was exclusively fed with breast milk was counted. In accordance with WHO recommendations, women who exclusively breastfed for at least six months were classified as exclusive breastfeeding (EBF), all others were classified as non-exclusive breastfeeding (mixed or formula feeding). The duration of exclusive breastfeeding was recorded in months. Infants who started complementary feeding before six months were considered non-exclusively breastfed.

### 2.8. Barriers and Facilitators of Exclusive Breastfeeding

To identify the barriers to EBF, we used a tool consisting of a series of statements designed to collect information on factors that may impede exclusive breastfeeding [24]. The main barriers identified through this tool included reasons related to work, lack of time, breast pain, self-perception of low milk production, maternal illnesses and medication use, fatigue, baby’s illness, excessive crying of the baby, and contradictory advice from family or friends. The instrument had a Cronbach’s alpha of 0.83 [24].

The tool also contained a series of statements related to commonly reported facilitators, such as personal conviction, family support, adequate rest, knowledge of the benefits of breastfeeding, having access to a private space for milk expression, availability of time, good maternal nutrition, advice from healthcare professionals, and perseverance.

Participants were asked to respond to each statement about barriers and facilitators with “yes” or “no”, indicating whether they perceived the factor as a barrier or a facilitator in their own experience. The tool consisted of 26 questions with two answer options: yes or no (9 questions related to barriers and 17 to facilitators).

### 2.9. Assessment of Food Insecurity

To assess food insecurity (FI) in the participants’ households, we used the Mexican Food Security Scale. This scale consists of twelve questions directed to the baby’s mother, with “yes” or “no” response options. Each “yes” response received a score of 1, while a “no” response received a score of 0. Based on the total score, households were classified as food insecure if they obtained more than 1 point (regardless of the presence of minors in the household) [25].

### 2.10. Clinical and Neonatal Outcomes

The mode of delivery was recorded as vaginal or cesarean section. The number of pregnancies was also recorded, and these data were classified as primiparous and multiparous. Birth weight and length were obtained via the standardized technique, using a SECA 374 “Baby and Mommy” scale (0.1 g accuracy) and a SECA 416 stadiometer (0.1 cm accuracy), and were recorded in the medical history. Subsequently, this information was categorized, according to the Intergrowth-21 curves, as small for gestational age (SGA), appropriate for gestational age (AGA), or large for gestational age (LGA) [26]. Other conditions of the baby at birth were also recorded: prematurity, macrosomia, heart or intestinal defects, and diseases of the respiratory system.

### 2.11. Sample Size

The sample size was calculated based on an expected frequency of EBF of 50% in women with high-risk pregnancies (given that the actual frequency in Mexico was unknown), with a confidence of 95% and an accuracy of 80%, for a finite population of 4000 live births at INPer during the two years prior to data collection. A non-probabilistic sample was selected. For the sample size calculation, the Open EPI^®^ calculator was used ([EDFFNp (1 − p)]/[(d2/Z21 − α/2(N − 1) + p×(1 − p)]), obtaining an estimated size of 422 cases. At the end of the investigation, the statistical power was determined with the G-Power^®^ program for studies of difference of proportions, with an OR of 1.794 from the crude model, a proportion of 55% of women with a category of high-risk pregnancy prior to conception and 71% with previous conditions + obstetric complications in pregnancy, a critical Z value of 3.2574, and a statistical power of 0.99% for the achieved sample size (566 women).

### 2.12. Statistical Analysis

According to the nature of the variables, we calculated means and standard deviations or medians with interquartile ranges, as appropriate. For categorical variables, we calculated frequencies (percentages). Subsequently, we examined the sociodemographic, clinical, and obstetric characteristics, as well as the barriers and facilitators, in relation to breastfeeding type (exclusive vs. non-exclusive). We converted the categorical variables into dummy variables for use in multivariate logistic regression models.

These models aimed to identify the association of prepregnancy conditions, pregnancy complications, food insecurity (FI), barriers, and facilitators with non-exclusive breastfeeding. The following exposure variables were used to perform four multivariate models: (1) Model 1 included high-risk prepregnancy conditions and pregnancy complications; (2) Model 2 included the variables from Model 1 plus FI; (3) Model 3 included the variables from Model 2 plus the barriers; and (4) Model 4 included the variables from Model 3 plus the facilitators. The outcome variable was non-exclusive breastfeeding.

All models are presented with the odds ratio (OR) and 95% confidence interval (CI).

All information was collected and analyzed using SPSS v. 23 software for Windows (SPSS Inc., IBM Corp., Armonk, NY, USA). A *p*-value ≤ 0.05 indicates statistical significance.

### 2.13. Ethical Considerations

The present study was carried out in strict compliance with the ethical principles established in the Declaration of Helsinki for research in human beings. The research protocol was reviewed and approved by the Research, Ethics, and Biosafety Committees of INPer, Mexican Ministry of Health (Registration number: 2020-1-15; approval date: 20 December 2020).

Participation in the study was voluntary and preceded by obtaining written informed consent from all participants. In the case of adolescent mothers, in addition to the participant’s informed consent, informed assent was obtained from their legal guardians, in the presence of two witnesses who certified the process. Invitations to participate were made in a private environment within the obstetrics ward of INPer, ensuring the absence of coercion.

To safeguard the confidentiality and privacy of the participants, the following data protection measures were implemented: anonymization, restricted access, and confidentiality. These measures guaranteed the protection of the rights and well-being of the participants, in accordance with applicable ethical and legal norms.

At our institute, newborns and women requiring medical, psychological, or social care are offered follow-up care as part of routine practice. Specifically, regarding breastfeeding support, all pregnant women receive two 90 min online sessions during the prenatal stage: (1) a discussion of the benefits of breastfeeding for both mother and infant, and (2) one week later, instruction on different breastfeeding positions using a doll as a model. After delivery, they receive individualized daily counseling on optimal breastfeeding positions until discharge. If the baby or mother is hospitalized, they also receive instructions on milk expression, with personalized guidance from breastfeeding educators.

## 3. Results

Six hundred and ninety-six participants were invited to participate and signed informed consent. For adolescent participants, both the adolescent and their parents or guardians signed the informed consent. At birth, 664 participants remained in the study; a total of 32 were excluded due to lack of time or interest in continuing. Postpartum, 65 additional participants were excluded due to fetal death, missing information, or delivery occurring at a hospital other than the Instituto Nacional de Perinatología. By the second interview, 566 participants remained, with 33 not attending. The overall participant loss was 18.7%. No significant differences were found between the participant group (*n* = 566) and the non-participant group (*n* = 130) (see Appendix A and Figure 2).

This study included 566 women, with a mean age of 29 ± 2.9 years. Of the sample, 12% were under 20 years old, 59% were between 20 and 34 years old, and 28% were over 35 years old. The median duration of breastfeeding was 4 months (interquartile range: 1–6). Consequently, the average frequency of non-exclusive breastfeeding was 56.4%. The results presented in Table 1 indicate that a higher frequency of exclusive breastfeeding was significantly related (*p* < 0.050) to the following characteristics: younger age, being a stay-at-home mother, lower educational level, and some level of FI. Furthermore, women who practiced exclusive breastfeeding initiated complementary feeding more frequently after 6 months (*p* < 0.001). In cases where complementary feeding was initiated before 6 months, it was carried out between 3 and 5 months of age and represented 21.2% of the infants. Furthermore, 78.8% of the infants initiated complementary feeding after 6 months of age.

Figure 3 shows the variables related with the duration of EBF. An inverse relationship was observed between the duration of EBF and the following variables: the number of perceived barriers, the number of high-risk conditions during pregnancy, maternal age, and the number of children (red line). In contrast, a direct relationship was found between the duration of EBF and the FI score (green line).

Among the clinical and obstetric characteristics, it was observed that primiparous women (*p* = 0.001), those who were overweight or obese (*p* = 0.030), and those with a high-risk prepregnancy condition (*p* = 0.001) or a high-risk prepregnancy and pregnancy condition (*p* = 0.005) had a higher frequency of non-exclusive breastfeeding. In contrast, pregnancy complications (*p* = 0.143) and newborn complications (*p* = 0.151) showed no significant differences in frequency based on breastfeeding practices.

Breastfeeding prevalence: Of the 566 participants, 312 had prepregnancy risk factors. Among these 312, 232 experienced both prepregnancy conditions and complications during pregnancy, childbirth, or the puerperium; only 201 (35.5%) reported exclusive breastfeeding for at least six months. Regarding gestational risks (complications), 406 participants experienced complications during pregnancy, and of these, 232 also had prepregnancy conditions. Among these 232, 171 (30.2%) reported exclusive breastfeeding for at least six months, while 235 (57.9%) reported not doing so exclusively. A total of 486 participants had either prepregnancy or gestational risk factors, of whom 80 had prepregnancy conditions only, and 174 had complications during pregnancy, childbirth, and the puerperium only. Of these 486 participants, 201 (41.4%) reported exclusive breastfeeding for at least six months, and 285 (58.6%) reported not having breastfed exclusively. In total, 80 participants had normal pregnancies; of these, 46 (57.5%) reported exclusive breastfeeding (see Table 2 and Appendix A).

Breastfeeding Prevalence According to Complications Presented by Newborns.

In our sample, 2 cases (0.4%) presented spina bifida, 160 (28.3%) prematurity, 2 (0.4%) cleft palate or cleft lip, 2 (0.4%) neonatal heart failure, and 58 (10.2%) various congenital defects. Due to the small sample size for each individual complication, these conditions were analyzed as a combined variable: presence or absence of newborn complications. No statistical significance was found according to breastfeeding practice.

Table 3 shows that non-exclusive breastfeeding was more frequent in women who reported having faced some barrier to breastfeeding. In contrast, EBF was more frequent in participants who reported having facilitators such as a good latch and positioning of the baby (*p* = 0.033), patience and perseverance (*p* = 0.005), availability of time (*p* = 0.034), counseling from health personnel (*p* = 0.018), conviction and knowledge about breastfeeding (*p* = 0.020), adequate rest (*p* = 0.022), and having someone to feed the baby when the mother works outside the home (*p* = 0.012).

The four regression models revealed that pre-existing high-risk conditions and multiparity were associated with a higher risk of not breastfeeding. In model M2, the incorporation of FI showed a protective effect on non-exclusive breastfeeding. In model M3, it was highlighted that the presence of facilitators such as good nutrition, counseling from health personnel, and access to adequate spaces for breastfeeding or milk expression was associated with a lower risk of non-exclusive breastfeeding. Model M4, which included additional factors, demonstrated that women with paid employment outside the home, a self-perception of low milk production, presence of illness or medication use, and a feeling of fatigue showed a higher probability of not breastfeeding exclusively, as presented in Table 4.

This study did not find significant bivariate relationships between FI and sociodemographic variables, nor between FI and the presence of pre-existing or pregnancy complications in the studied population. Furthermore, gestational and neonatal complications were not associated with EBF.

## 4. Discussion

The present study showed that pre-existing pathological conditions are a determining factor for the failure of EBF. This finding highlights the importance of optimizing maternal health before conception to promote successful breastfeeding. However, it is important to consider that multiple factors can influence breastfeeding outcomes. Other variables negatively influence exclusive breastfeeding practice. Maternal employment, especially in work environments that do not facilitate milk expression and storage, represents a significant barrier. Likewise, a self-perception of low milk production, although not always based on reality, can lead to unnecessary supplementation and premature abandonment of breastfeeding. The use of certain medications, as well as various maternal health conditions (such as postpartum depression, infections, or chronic diseases), also increase the risk of non-exclusive breastfeeding. Future research should explore in greater detail the interaction between these factors and develop specific interventions to promote exclusive breastfeeding in women with pre-existing health conditions or who face particular challenges.

### 4.1. Breastfeeding Practice in Women with High-Risk Pregnancies

Women with health problems before or during pregnancy tend to initiate and maintain breastfeeding less frequently than healthy women, as demonstrated in our study. These health problems, which may require specialized obstetric care, increase the risk of complications during pregnancy and childbirth. Factors such as gestational diabetes, high blood pressure during pregnancy, chronic diseases (such as inflammatory bowel disease or epilepsy), obesity, and smoking have been linked to a shorter duration of breastfeeding [27,28,29,30]. For example, the study by Kozhimannil KB et al. [31] showed that women with complications during pregnancy (such as taking medication for blood pressure, having diabetes, or a high BMI) were less likely to exclusively breastfeed their babies at one week postpartum. Another study showed that maternal chronic diseases did not affect the initiation or continuation of breastfeeding up to 6 months but increased the likelihood of discontinuing exclusive breastfeeding before that time [29]. Furthermore, smoking during pregnancy is associated with a lower probability of initiating breastfeeding and a shorter duration of breastfeeding [30].

Moreover, the exact reason why mothers with a history of diabetes tend to breastfeed exclusively for a shorter time is unclear. Research suggests that diabetes and metabolic problems may delay milk production [32]. Given that obesity increases the risk of gestational diabetes, mothers with pre-existing diabetes may also experience this delay [33]. In addition, giving supplements to the baby could decrease breast milk production in these mothers and, therefore, reduce breastfeeding rates [34]. Some diabetic women may also have trouble extracting colostrum in the beginning [35].

Our results differ partially from what was reported by Scime NV et al. [36] in Canada, who demonstrated a high prevalence of breastfeeding initiation because most mothers start by breastfeeding their babies, possibly due to the excellent support they receive in hospitals [37]. This could explain why no differences were observed in breastfeeding initiation based on prebirth medical risk. The conflicting information about breastfeeding initiation and prenatal medical risk may be due to differences in hospital policies and cultural norms regarding infant feeding. However, once mothers begin breastfeeding, prenatal medical risk seems to hinder the continuation of breastfeeding, regardless of demographic factors or complications in childbirth. This risk could affect breastfeeding at a biological level, such as the delay in milk production in women with diabetes, obesity, or alcohol consumption during pregnancy, which can shorten the duration of breastfeeding [38,39]. Differences in milk composition have also been observed, such as immune factors, in women with pre-eclampsia or smokers, suggesting an impact on mammary function [40].

In our study, the bivariate correlation analysis with continuous variables (Figure 3) revealed generally low correlations between the predictors of breastfeeding practices, except for pre-pregnancy risk. This finding could explain the lack of statistical significance of regression models with breastfeeding as the dependent variable. Similarly, examining the same categorized variables (using dummy variables) yielded consistent results. These findings suggest that the absence of association is not attributed to the coding of the variables, which reinforces the robustness of our results.

More research is needed to understand how prenatal medical history can interfere with breastfeeding. The experiences of women are also important, as accounts from women with complications in pregnancy or chronic diseases highlight the prolonged physical recovery after childbirth and concern about the safety of medications during breastfeeding [41].

### 4.2. Food Insecurity

Our results reveal a frequency of FI lower than that documented in national studies conducted in Mexico more than a decade ago, using a representative sample [42]. However, we observed that participants who experienced food insecurity showed a higher prevalence of EBF. This finding contrasts with a study conducted in the United States in women without complications, where food insecurity was not significantly associated with the practice of EBF [43]. It also differs from another investigation in which women with food insecurity received food support, and in which FI was associated with a higher frequency of non-exclusive breastfeeding [44].

In the context of our study, it is possible that participants with food insecurity received comprehensive advice on the benefits of EBF, possibly from the prenatal stage, from trained medical and paramedical personnel. This intervention could have positively influenced their decision to breastfeed. It is relevant to note that the INPer, where the study was conducted, is accredited as a “Baby and Woman Friendly Hospital”. This designation implies the periodic certification of staff to ensure they are up-to-date on the best practices related to the care of pregnant women. However, it is important to recognize that, in contexts such as high-risk pregnancy, significant barriers remain that hinder the successful establishment of exclusive breastfeeding.

### 4.3. Barriers and Facilitators of Non-Exclusive Breastfeeding

One of the most frequently cited barriers to breastfeeding in various studies is maternal employment, similar to the findings of our research. In formal and informal workplaces, the lack of conditions, spaces, and facilities to maintain exclusive breastfeeding has prevailed despite scientific reports demonstrating the need for such conditions for successful EBF among mothers, even more so in informal work sectors, where the majority of our participants came from [45,46,47,48,49]. Therefore, it is important to document that given a global landscape of obesity like that of many low- and middle-income countries, and the limited practice of exclusive breastfeeding in women who experienced high-risk pregnancies, the quality of life of the mother–baby dyad can be undermined, with high costs for the health sector and for the family [18].

Furthermore, our study showed that the self-perception of low human milk production and maternal fatigue were risk factors for failing to achieve successful exclusive breastfeeding. The self-perception of low milk production has been related to opinions from family and/or the partner, but especially to the mother’s confidence in breast milk production, which increases according to the baby’s needs, as demonstrated by Rodrigo R et al. [50] in a group of mothers originating from Sri Lanka, who concluded that greater confidence leads to greater milk production, which promotes tranquility in the mother and therefore uninhibited milk production.

It is also necessary to mention the role of health personnel in Baby and Woman Friendly Hospitals, such as INPer, where pregnant women and mothers are encouraged from the prenatal stage to breastfeed their babies. Despite this, our study did not reach the figures for EBF that are recommended by the WHO. The failure to reach a higher proportion of mothers practicing EBF is probably due to the fact that the group of pregnant women had pre-existing diseases or complications during pregnancy, which made it impossible for them to practice EBF in some cases, or in others, it was due to a lack of knowledge about the practice of breastfeeding in women who experienced a high-risk pregnancy, as documented in the study by Kang et al. [51], who demonstrated that a high-risk pregnancy due to pregestational health conditions can affect the mother’s safety and confidence in adequate breast milk production, coupled with various physiological processes derived from the excess weight of some mothers as well as other health conditions that can affect lactogenesis II.

In our study, we observed that the perception of certain barriers and facilitators were associated with breastfeeding practice in various ways. Hence, the continuous training for all personnel in maternal–infant hospitals is of great importance, for the purpose of guiding and helping resolve all the doubts expressed by women who are experiencing a high-risk pregnancy, as well as their families and partners, always providing truthful information that answers concerns and makes clear the benefits of EBF for both mother and baby. In addition, having adequate spaces for milk expression and empathic health personnel in relation to mothers with hospitalized babies or babies who are separated from their mother due to some maternal complication is also of great importance [52,53].

In the present study, we observed that the pregestational conditions that define a high-risk pregnancy were associated with a higher probability of practicing non-exclusive breastfeeding, for example, due to obesity, arterial hypertension, type 1 or 2 diabetes mellitus, and HIV, among others, which occur more frequently in women over 35 years of age or multigravidas [12,54,55]. All these conditions can occur interrelated or separately; however, contrary to what we expected, multiparity was associated with a higher risk of practicing non-exclusive breastfeeding. This situation can be attributed to the fact that with a greater number of children, there is a greater age, and women probably already have a job outside the home, which does not allow them to breastfeed the baby on demand or in optimal conditions. In addition, there may be some physiological changes due to different diseases prior to pregnancy that can influence a low secretion of prolactin and consequently a lower milk production, for example, this situation occurs in the case of mothers with an excessive amount of adipose tissue, although the self-efficacy that the mother presents also has a determining role in the production of breast milk [56].

With regard to women who were dedicated to the home, those under 19 years of age, and with less schooling, showed a higher frequency of EBF. This could be explained by the fact that younger mothers usually have a lower level of schooling, a lower body mass index, and usually dedicate themselves to household activities; therefore, they have more time available for breastfeeding on demand. This indicates that working conditions outside the home in several countries limit EBF and promote the initiation of early complementary feeding, as observed in our study [57].

### 4.4. Limitations

Our study has several limitations that should be considered when interpreting the results.

First, the cross-sectional design of the study makes it difficult to establish causal relationships between the factors analyzed and the observed breastfeeding outcomes. Despite this limitation, this research, to our knowledge, is the first in Mexico to analyze the factors associated with non-exclusive breastfeeding. Specifically, it examines barriers, facilitators, and food insecurity in mothers with high-risk prepregnancy conditions, adverse pregnancy outcomes, and neonatal complications. This innovative approach generates new research questions that, in the long term, could contribute to the design of programs aimed at promoting healthy lifestyles before pregnancy.

Second, the information provided by mothers could be inaccurate due to potential difficulties in recalling precise details about their breastfeeding practices during the first six months. This recall bias is common in retrospective studies and could have affected the accuracy of the results.

Third, the sample selection method, based on convenience rather than random sampling, could have resulted in a sample that does not adequately represent all social classes.

Fourth, another limitation is the lack of newborn follow-up, which could have led to an underestimation of breastfeeding-related problems.

These limitations restrict the generalizability of the results to the broader population, as the sample may be biased toward certain demographic groups.

Despite these limitations, our study underscores the importance of early interventions to improve maternal health and promote successful breastfeeding, given the predominance of pregestational risk conditions associated with the practice of non-exclusive breastfeeding.

### 4.5. Clinical Implications

One of the most relevant clinical implications of our research is that it reiterates the critical importance of starting a pregnancy with an optimal pregestational nutritional status. The findings underscore that a high-risk pregnancy not only complicates perinatal outcomes but also negatively impacts the practice of EBF. This highlights the need for early interventions that promote maternal health before conception, including strategies to improve nutrition and reduce modifiable risk factors.

Furthermore, our research reaffirms the relevance of the information and support that health personnel provide to mothers and their families in relation to the decision to breastfeed. It is essential that health professionals are trained to provide accurate and up-to-date information on the benefits of EBF, as well as to address concerns and challenges that may arise during the breastfeeding process.

Likewise, the urgency for workplaces to implement policies and create adequate and dignified spaces that allow for breast milk expression is highlighted. These measures are essential to facilitate the maintenance of EBF in working mothers, who often face difficulties in reconciling their work responsibilities with breastfeeding. The availability of milk expression spaces, along with policies that promote work flexibility and support for breastfeeding, can make a significant difference in the duration and success of EBF.

### 4.6. Perspectives

Various factors influence a woman’s decision to breastfeed and her success in lactation, including the support of health professionals and family, cultural customs, and social conditions that impact her health and well-being. Hospitals play a fundamental role in promoting breastfeeding through initiatives such as the Baby Friendly Hospital Initiative, which promotes EBF and restricts the use of formula to medically justified situations. In the context of care for women with diabetes, prenatal colostrum harvesting can be a valuable strategy. After delivery, it is crucial to facilitate immediate skin-to-skin contact, initiate early breastfeeding, respond to the baby’s hunger cues, instruct mothers in effective breastfeeding techniques, and encourage the rooming-in of the mother and baby. In addition, it is important to emphasize that the continuity of support and education after hospital discharge is essential to maintain and consolidate breastfeeding.

Furthermore, the consumption of certain drugs, such as some antidepressants or medications that decrease milk production, and the presence of maternal health problems, such as postpartum depression, infections, or chronic conditions, increase the likelihood of not breastfeeding. It is suggested that subsequent studies delve into how these factors relate to each other and design specific strategies to promote exclusive breastfeeding in women with pre-existing health problems or who face specific challenges.

## 5. Conclusions

The research revealed that the failure of exclusive breastfeeding is significantly influenced by the presence of pre-existing pathological conditions in the mother. This discovery underscores the need to improve the health of mothers before pregnancy to facilitate successful lactation. Nevertheless, it is crucial to recognize that breastfeeding is affected by a variety of factors.

In addition to pre-existing conditions, several factors can hinder the practice of exclusive breastfeeding. Among these, the mother’s work, particularly in environments not conducive to milk extraction and storage, constitutes a considerable obstacle. Furthermore, the belief of mothers that they produce little milk, even if it is not always true, can lead to unnecessarily supplementing the baby’s diet and interrupting breastfeeding prematurely.

## Figures and Tables

**Figure 1 nutrients-17-02309-f001:**
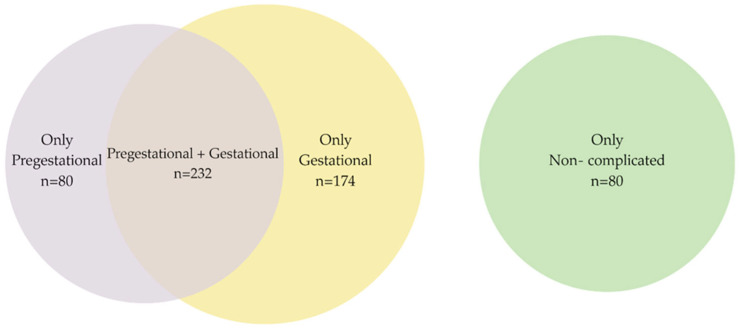
**Venn diagram.** Pre- and gestational conditions.

**Figure 2 nutrients-17-02309-f002:**
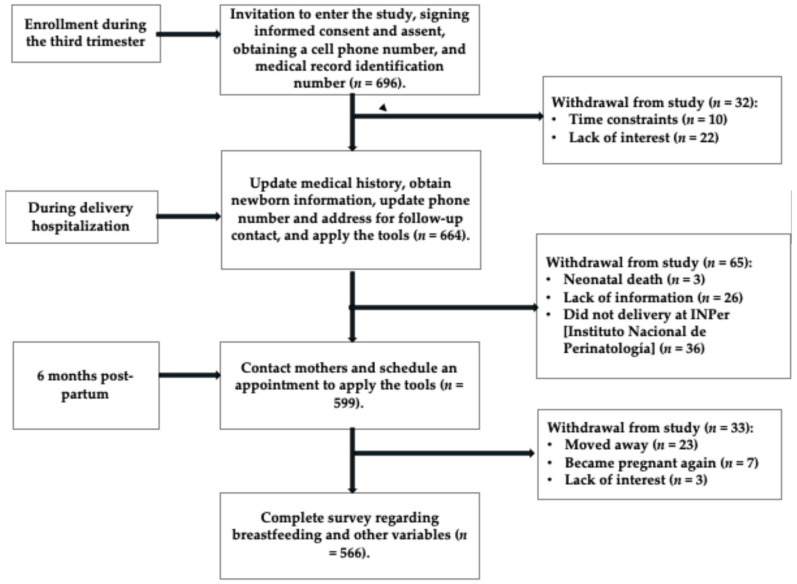
Flow diagram, Illustration of recruitment, exclusions, and the final study sample size.

**Figure 3 nutrients-17-02309-f003:**
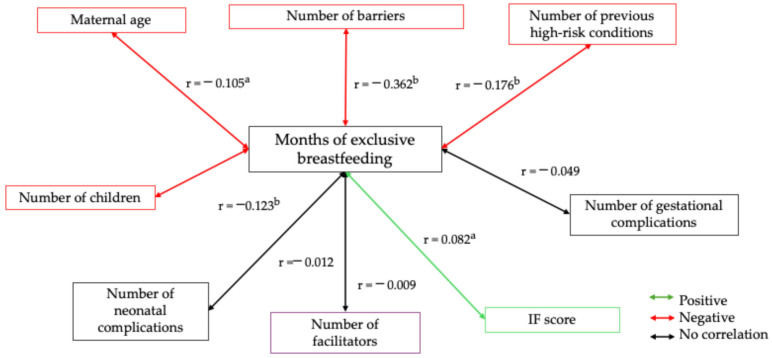
Spearman’s correlation of months of exclusive breastfeeding with maternal age, parity, and socioeconomic status. Spearman’s bivariate correlation test. [(a) *p* ≤ 0.050; (b) *p* ≤ 0.001]. Positive correlation (green line), negative correlation (red line), and no correlation (black line).

**Table 1 nutrients-17-02309-t001:** Sociodemographic characteristics of the sample, by breastfeeding status, *n* (%).

Variables		Exclusive Breastfeeding, *n* = 247	Non-Exclusive, *n* = 319	* *p*
Age category	<20 y, *n* = 68	39 (57.4)	29 (42.7)	0.024
	20–34 y, *n* = 339	148 (43.7)	191 (56.3)	
	≥35 y, *n* = 159	60 (43.7)	99 (56.3)	
Occupation				
	Home, *n* = 411	189 (46.0)	222 (54)	0.041
	Worker, *n* = 155	58 (37.4)	97 (62.6)	
Economic support				
	Herself, *n* = 77	35 (45.5)	42 (54.5)	0.826
	Partner, *n* = 370	158 (42.7)	212 (57.3)	
	Parents, *n* = 119	54 (45.4)	65 (54.6)	
Civil status				
	Married, *n* = 396	169 (42.7)	227 (57.3)	0.270
	Single, *n* = 170	78 (45.9)	92 (54.1)	
Socioeconomic level				
	Very low, *n* = 185	83 (44.9)	102 (55.1)	
	Low, *n* = 249	102 (41.0)	147 (59.0)	0.488
	Middle, *n* = 132	62 (47.0)	70 (53.0)	
Educational level				
	Secondary and less, *n* = 199	89 (44.7)	110 (55.3)	0.011
	High school, *n* = 209	104 (49.8)	105 (50.2)	
	University, *n* = 158	54 (34.2)	104 (65.8)	
Food insecurity				
	None, *n* = 507	214 (42.2)	293 (57.8)	0.031
	Moderate and severe, *n* = 59	33 (55.9)	26 (56.4)	
Starting of complementary feeding (months)				
	<6	26 (10.5)	94 (29.5)	0.001
	≥6	221 (89.5)	225 (70.5)	

* *p*-value of Pearson’s Chi-square.

**Table 2 nutrients-17-02309-t002:** Clinical and obstetric characteristics by breastfeeding type, *n* (%).

Variables	Breastfeeding		* *p*
	Exclusive, *n* = 247 (43.6%)	Non-Exclusive, *n* = 319 (56.4%)	
Parity			
Primiparous	186 (48.2)	200 (51.8)	0.001
Multiparous	61 (33.9)	119 (66.1)	
Pregestational body mass index			
Low weight	10 (1.8)	9 (1.6)	0.030
Normal weight	107 (18.9)	122 (21.6)	
Overweight	84 (14.8)	103 (18.2)	
Obesity	46 (8.1)	85 (15.0)	
Gestational weight gain			
Insufficient	95 (16.8)	136 (24.0)	0.580
Adequate	56 (10.0)	64 (11.4)	
Excessive	95 (16.8)	119 (21.0)	
Delivery			
Cesarean section	120 (21.2)	170 (30.0)	0.152
Vaginal delivery	127 (22.4)	149 (26.4)	
Pregestational high risk			
No	131 (23.1)	123 (21.8)	0.001
Yes	116 (20.5)	196 (34.6)	
Gestational risk (complications)			
No	76 (13.4)	84 (14.8)	0.245
Yes	171 (30.2)	235 (41.6)	
Pregestational and gestational high risk			
No	46 (8.1)	34 (6.0)	0.005
Yes	201 (35.5)	285 (50.4)	
Neonate			
Gestational age at birth			
Very premature	22 (37.0)	35 (63.0)	0.795
Moderate premature	60 (57.6)	72 (74.4)	
Non-premature (term)	165 (43.8)	212 (56.2)	
Neonate complications			
No	102 (18.0)	117 (20.7)	0.151
Yes	145 (25.6)	202 (35.7)	
Weight for gestational age			
SGA	68 (12.0)	94 (16.6)	0.867
AGA	163 (28.8)	206 (36.4)	
LGA	16 (2.8)	19 (3.4)	

* *p*-value of Pearson’s Chi-square. SGA: small for gestational age; AGA: adequate for gestational age; LGA: large for gestational age.

**Table 3 nutrients-17-02309-t003:** Type of breastfeeding by reported barriers and facilitators to breastfeeding, *n* (%).

Variable	Breastfeeding		** p*
	Exclusive, *n* = 247	Non-Exclusive, *n* = 319	
Barriers			
Working	6 (16.7)	30 (83.3)	0.001
Self-perception of low milk production	9 (10.1)	80 (89.9)	0.001
Maternal diseases and their medications	5 (20.0)	20 (80.0)	0.011
Lack of time	4 (13.8)	25 (86.2)	0.001
I feel tired	3 (10.3)	26 (89.7)	0.001
Because of my baby’s illness	3 (16.7)	15 (83.3)	0.019
My baby cries a lot	9 (14.8)	52 (85.2)	0.001
My family advised me to use formula	7 (18.9)	30 (81.1)	0.001
I feel pain in my breast	9 (17.3)	43 (82.7)	0.001
Facilitators			
Good position and accommodation	110 (39.6)	168 (60.4)	0.033
Have patience and perseverance	136 (49.6)	138 (50.4)	0.005
Good nutrition	161 (42.9)	214 (57.1)	0.350
Availability of time	114 (39.2)	177 (60.8)	0.034
Advice from healthcare personnel	92 (50.3)	91 (49.7)	0.018
Partner and family support	137 (46.9)	155 (53.1)	0.062
Conviction and mastering the subject of breastfeeding	140 (47.9)	152 (52.1)	0.020
Good mood	138 (43.4)	180 (56.6)	0.481
Adequate resting of mother	185 (46.5)	213 (53.5)	0.022
Adequate resting of mother and baby	157 (44.5)	196 (55.5)	0.668
Adequate space to give breastfeeding	122 (43.7)	157 (56.3)	0.517
Have my baby from the first 4 h after birth	144 (44.4)	180 (55.6)	0.359
While I work, someone feeds my child breast milk.	132 (48.7)	139 (51.3)	0.012
A baby’s adequate resting	95 (42.0)	131 (58.0)	0.295
To recover the pregestational weight	146 (44.8)	180 (55.2)	0.290
To know how to preserve milk	119 (44.7)	147 (55.3)	0.341
Adequate space to express breast milk	122 (43.7)	157 (56.3)	0.157

* *p*-value of Pearson’s Chi-square.

**Table 4 nutrients-17-02309-t004:** Multivariable regression models predicting non-exclusive breastfeeding.

Variable	OR	CI 95%	*P*
M1 with risk and complications			
Pregestational high-risk conditions	1.794	1.280–2.516	0.003
Gestational risk conditions	0.490	0.781–1.678	0.490
Pregestational high-risk and gestational risk conditions	1.415	0.665–3.012	0.368
Neonate complications	1.197	0.840–1.706	0.321
Multiparity	1.687	1.162–2.451	0.006
M2 with Food insecurity			
Pregestational high-risk conditions	1.838	1.308–2.584	0.001
Gestational risk conditions	1.171	0.796–1.721	0.423
Pregestational high-risk and gestational risk conditions	1.365	0.639–2.916	0.422
Neonate complications	1.187	0.831–1.695	0.346
Multiparity	1.691	1.163–2.461	0.006
Food insecurity	0.530	0.303–0.927	0.026
M3 with facilitators			
Pregestational high-risk conditions	1.754	1.237–2.489	0.002
Gestational risk conditions	1.234	0.829–1.836	0.300
Pregestational high-risk and gestational risk conditions	1.269	0.582–2.771	0.549
Neonate complications	1.216	0.841–1.757	0.299
Multiparity	1.723	1.173–2.531	0.005
Food insecurity	0.531	0.301–0.938	0.029
Good positions	0.571	0.233–1.397	0.220
Patience and perseverance	1.067	0.433–2.629	0.887
Good nutrition	0.245	0.069–0.868	0.029
Availability time	1.090	0.444–2.678	0.821
Advice from healthcare personnel	0.120	0.032–0.440	0.001
Partner and family support	0.576	0.233–1.425	0.233
Have adequate space to feed the baby and express milk	0.288	0.085–0.974	0.045
M4 with barriers			
Pregestational high-risk conditions	1.564	1.063–2.301	0.032
Gestational risk conditions	1.273	0.822–1.970	0.279
Pregestational high-risk conditions and gestational risk	1.505	0.636–3.562	0.352
Neonate complications	1.136	0.959–2.151	0.079
Multiparity	1.643	1.081–2.496	0.020
Food insecurity	0.405	0.209–0.784	0.007
Good positions	0.468	0.151–1.445	0.187
Patience and perseverance	0.781	0.250–2.438	0.671
Good nutrition	0.175	0.035–0.875	0.034
Availability time	0.834	0.289–2.404	0.736
Advice from healthcare personnel	0.096	0.018–0.516	0.006
Partner and family support	0.856	0.297–2.471	0.774
Have adequate space to feed the baby and express the milk	0.327	0.066–1.625	0.172
Works	4.681	1.658–13.214	0.004
Self-perception of low milk production	6.453	2.953–14.102	0.001
Maternal diseases and their medications	3.918	1.361–11.283	0.011
Lack of time	1.770	0.509–6.157	0.369
I feel tired	4.670	1.153–18.914	0.031
Because of my baby’s illness	1.671	0.389–7.173	0.490
My baby cries a lot	1.594	0.640–3.969	0.316
My family advised me to use formula	1.129	0.393–3.244	0.822
I feel pain in my breast	2.190	0.902–5.313	0.083

## Data Availability

The data presented in this study are available from the corresponding authors upon reasonable request.

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
