# Peer review of "Association of Pre- and Gestational Conditions and Barriers to Breastfeeding with Exclusive Breastfeeding Practices"

_nutrients, 2025, doi:10.3390/nu17142309_

Round 1
Reviewer 1 Report
Comments and Suggestions for Authors
The manuscript entitled “Pre-pregnancy conditions and barriers to breastfeeding in-crease the risk of not breastfeeding more than obstetric complications” presents interesting issues however some questions arise
- The title is not precise, as it suggests a certain causality, while the study is cross-sectional. Perhaps this title would be better: Association of pre-pregnancy conditions and breastfeeding barriers with exclusive breastfeeding practices. This is just a suggestion for the title; authors may propose another one that better reflects the nature and results of the study.
- Information regarding recruitment should be included - specifically, how many women were approached to participate in the study, what percentage agreed to take part, and how many declined. It is necessary to include an assessment of the risk of selection bias (or discussed).
- The questionnaire needs to be described precisely. If a Cronbach’s alpha analysis was conducted, this indicates that it was a multi-item tool (how many?) designed to assess a specific construct (e.g., an attitude or perception), and this should be clearly stated. Additionally, for the remaining questions, it should be clarified whether they were inspired by an existing questionnaire.
- Was the tool validated? Please provide information regarding its reliability and validity.
- “Age was obtained through direct questioning and categorized according to obstetric risk into three groups: ≤19 years, 20 to 34.9 years, and ≥35 years.” – Were there any underage participants (minors) included? This needs to be clarified, as in such cases, consent for participation in the study should be obtained from their parents or legal guardians.
- In the methodology section, more information should be provided about the Multivariable Regression (OR, CI, etc.).
- Table 1: “Starting of complementary feeding (months) ≤5.9” - Please specify from what age complementary feeding was initiated, as the current wording suggests that some participants may have started complementary feeding from birth.
- Table 1: *p-value by Pearson Chi-square. - Was Yates' correction applied in the 2x2 tables? If so, please include this information. If not, please provide the Yates' chi-square.
- The conclusions at the end of the article are very general and read more like a summary or a general statement about the positive effects of breastfeeding. The conclusions should be directly related to the results obtained during the analysis.
Author Response
First Reviewer
We would like to express our sincere gratitude to Reviewer 1 for their thorough and valuable review, which has significantly contributed to enriching our manuscript. Additionally, we deeply appreciate the time and effort dedicated to conducting this review. Below, we present detailed responses to each of the raised comments.
The manuscript entitled “Pre-pregnancy conditions and barriers to breastfeeding increase the risk of not breastfeeding more than obstetric complications” presents interesting issues however, some questions arise
The title is not precise, as it suggests a certain causality, while the study is cross-sectional. Perhaps this title would be better: Association of pre-pregnancy conditions and breastfeeding barriers with exclusive breastfeeding practices. This is just a suggestion for the title; authors may propose another one that better reflects the nature and results of the study.
Response: The suggestion is accepted, and consequently, the title of the manuscript has been modified.
Information regarding recruitment should be included - specifically, how many women were approached to participate in the study, what percentage agreed to take part, and how many declined. It is necessary to include an assessment of the risk of selection bias (or discussed).
Response: In accordance with the observation, a flow chart was included specifying the number of participants invited to participate and how many completed the study. Furthermore, a comparative statistical analysis was provided between the group that participated and the group that did not participate, demonstrating that there were no significant differences between the two groups. This information is shown in the flow chart and in supplementary table 1.
The questionnaire needs to be described precisely. If a Cronbach’s alpha analysis was conducted, this indicates that it was a multi-item tool (how many?) designed to assess a specific construct (e.g., an attitude or perception), and this should be clearly stated. Additionally, for the remaining questions, it should be clarified whether they were inspired by an existing questionnaire.
Response: In accordance with your observation, the questionnaire used to obtain the information has been described in greater detail. This questionnaire had already been validated and previously used by our research group [Sámano, R. and Chico-Barba, G., Armenteros-Martínez, T., Escamilla-Fonseca, N., Piélago-Álvarez, C. Aguilar-Álvarez, J. and Méndez-Celayo, S. Barriers and facilitators of exclusive breastfeeding practice in mothers from Mexico City. Archivos Latinoamericanos de Nutrición. 2018; 68(1):41–50. https://doi.org/10.37527/2018.68.1.004].
Was the tool validated? Please provide information regarding its reliability and validity.
Response: The reliability of the questionnaire was evaluated using Cronbach's alpha coefficient, yielding a value of 0.830, which indicates adequate internal consistency. The content validity was also evaluated using the agreement index (agreements out of the total responses), with a value of 0.87. This questionnaire was previously evaluated by our research group.
“Age was obtained through direct questioning and categorized according to obstetric risk into three groups: ≤19 years, 20 to 34.9 years, and ≥35 years.” – Were there any underage participants (minors) included? This needs to be clarified, as in such cases; consent for participation in the study should be obtained from their parents or legal guardians.
Response: Pregnant women of reproductive age (15-49 years) were included. In the case of pregnant adolescents, informed consent was obtained, signed by their parents or guardians, as well as assent signed by the adolescent. All this information is detailed in section 2.13, "Ethical Considerations"
In the methodology section, more information should be provided about the Multivariable Regression (OR, CI, etc.).
Response: In response to the observation, additional information on the multivariable regression was provided in section 2.12, "Statistical Analysis".
Table 1: “Starting of complementary feeding (months) ≤5.9” - Please specify from what age complementary feeding was initiated, as the current wording suggests that some participants may have started complementary feeding from birth.
Response: In accordance with the observation, it was specified in the text that 21.2% of the infants studied initiated complementary feeding before 6 months. Furthermore, it was specified that the age range in which complementary feeding was initiated in this group of children was between 3 and 5 months.
Table 1: *p-value by Pearson Chi-square. - Was Yates' correction applied in the 2x2 tables? If so, please include this information. If not, please provide the Yates' chi-square.
Response: In accordance with your observation, in all cases where the p-value was obtained using the Pearson's chi-squared test, the necessary assumptions for the application of that statistical test were met. Therefore, it was not necessary to apply Yates' correction.
The conclusions at the end of the article are very general and read more like a summary or a general statement about the positive effects of breastfeeding. The conclusions should be directly related to the results obtained during the analysis.
Response: We appreciate your valuable observation. The previous text has been removed, and the main conclusions of the study have been drafted.

Reviewer 2 Report
Comments and Suggestions for Authors
General: This observational, single-site, cross-sectional study was designed to assess the effect of a variety of factors on breastfeeding. However, the title appears restricted to a comparison of only two of these factors (pre-pregnancy conditions vs. obstetric complications). The study identifies a number of factors that correlate with non-exclusive breastfeeding, but generalizability of the findings is both limited by a number of factors the authors identify as well as a number of others noted below, including some internal inconsistencies in population definition, limited information concerning neonatal factors, rather low correlations coefficients in Figure 1, and unclear follow-up rates. Other revisions are also needed. Including the title.
Introduction: This provides a brief review of literature indicating impact of breastfeeding on a variety of infant and maternal outcomes and also notes a number of factors that have been found to influence initiation and continuation of breastfeeding. This study was designed to address a relative gap in research on the relationship between high-risk pregnancies and breastfeeding outcomes, as well as to analyze the association between a variety of factors, including pre-pregnancy characteristics, on the practice of exclusive breastfeeding. In the second sentence, the cited reference (#2) does not support the benefits noted, and “overall health” is not defined. In the third sentence, the reference cited (#3), mainly cites information from animal (rat) studies, and only supports the benefits for humans to a limited extent.
Materials and Methods:
- The inclusion criteria indicate this study is comprised “women with high-risk pregnancies,” which includes one or more of a specific list of conditions. However, Table 2 indicates that only 55% the study population are “pre-gestational height sic (presumably high)
- It is not clear what percent of birth population at this institution during the study timeframe was included or how representative the included population was of the broader population from which it was drawn. A consort diagram would be helpful.
- It is not clear what percent of women with one of these pre-pregnancy conditions, which were presumably identified during a third trimester visit, were actually seen at the 6 month post-partum follow-up visit, at which time information concerning the outcome variable, breastfeeding, was obtained. It would be unusual if 100% were seen at follow-up, which raises the question whether the sample was actually limited to those with follow-up, in which case, it would be important to know how those who didn’t follow-up compared with those who did.
- It is also not clear what “consecutive case sampling” refers to.
- The abstract refers to “children between 6 and 48 months;” but it is not clear what data was obtained after the six-month post-partum visit or how it was used.
- “Complicated” pregnancy was defined in terms of a set of conditions that developed during gestation. Mothers were classified into one of three groups, but does not appear from data in Table 2 that these were actually mutually exclusive groups, since the number of subjects in the combined “pregestational and gestational high risk groups” is less than the sum of the component groups, suggesting that there is overlap between the pre-gestational and gestational groups and that the combined category actually reflects pre-gestational and/or gestational high risk. This needs to be clarified.
- Though the birth hospital in this study is described as a “Bay and Woman Friendly” hospital, it is not clear what type of breastfeeding education and support women receive at this hospital, either prenatally during gestation, during birth admission, or on discharge.
- “Neonatal outcomes” in 2.10 appear to include “other conditions of the baby at birth,” though not all of the conditions listed would necessarily be apparent at birth, beyond weight for age and “prematurity” (which latter is not defined or further specified). Length of stay, age at initiation of oral feeding, or neurologic conditions are not included.
- No information was presented concerning the frequency/type of follow-up medical care the infants received, which could affect the ongoing support or lack thereof of breastfeeding; nor was any information provided concerning post-discharge illnesses that might have affected breastfeeding.
- Interestingly, the outcome variable chosen for the regression model analysis was “non-exclusive breastfeeding,” which would suggest that the benefits of breast-feeding are restricted to those who are exclusively breast-fed, which is not clear from this paper or other published literature. Also, since exclusive breastfeeding is defined as exclusive breastfeeding for at least six full months, this would further imply that exclusive breastfeeding for any shorter duration does not have any benefit. Also it is not clear whether introduction of beikost (solids) prior to six months likewise would be classified as non-exclusive breastfeeding, even if milk feeding is exclusively breast milk. The rationale for choosing non-exclusive vs. exclusive as an outcome variable needs to be explained.
- Though there are likely interactions among these variables, as briefly noted in the Discussion, no interactions are tested for in the regression analyses, though such is proposed for future research.
Results:
- Table 1 presents a chi-square comparison of exclusive breastfeeding (EBF) vs. non-exclusive across a spectrum of sociodemographic variables. Figure 1 shows Spearman’s Correlation between exclusive breastfeeding and a variety of factors that might contribute to this practice. Although 5 of 8 of these factors have statistically significant relationships, the correlation coefficients are all rather low.
- Table 2 shows chi-square correlations between various “medical” variables and breastfeeding practices. The categories in this table need to be left-aligned with consistent indentation of subcategories. As noted above, “pregestational height risk” needs to be changed to “pregestational high risk;” and “pregestational and gestational high risk” presumably should be “pregestational and/or gestational high risk.” “Neonate” presumably is equivalent to “baby” and only one of the two needs to be included.
- Table 3 presents chi-square correlations between various barriers and various facilitators and breastfeeding. As with Table 2, these items should be left-aligned.
- Table 4 shows the results of 4 progressively inclusive regression models in predicting non-exclusive breast feeding. As with other tables, left-alignment of the individual variables for each model would be easier to read. Though many of the statistically factors have some biological plausibility as noted in the Discussion, others are less intuitive, most notably “neonatal complications,” though that factor may be due to a restrictive definition of neonatal factors, as noted above.
- The title of the article focuses on a comparison of pre-pregnancy conditions with obstetric complications; however, this is not discussed as such in results either in relation to data in Table 2 or Table 4, though the data in Table 4 do show that the OR for high risk pre-pregnancy conditions is statistically significant in all four models whereas pregnancy complications is not.
Discussion/Conclusion:
- In 4.1 reportedly 43% of women with high risk-pregnancies practiced EBF; however, it is not clear how that number ties to related data in Table 2 or in the narrative in Results.
- As noted in #5 under results, some discussion concerning relative risk of pre-pregnancy conditions vs. obstetric complications is warranted. Even so, that is not the only, or even most salient, factor identified in this study. Moreover, that comparison is not included in the conclusion.
- The conclusion lists some of the possible ramifications of the findings of this study, but some of that is speculative or based on other studies.
Author Response
Second reviewer
We would like to express our sincere gratitude to Reviewer 2 for their thorough and valuable review, which has significantly contributed to enriching our manuscript. Additionally, we deeply appreciate the time and effort dedicated to conducting this review. Below, we present detailed responses to each of the raised comments.
General: This observational, single-site, cross-sectional study was designed to assess the effect of a variety of factors on breastfeeding. However, the title appears restricted to a comparison of only two of these factors (pre-pregnancy conditions vs. obstetric complications). The study identifies a number of factors that correlate with non-exclusive breastfeeding, but generalizability of the findings is both limited by a number of factors the authors identify as well as a number of others noted below, including some internal inconsistencies in population definition, limited information concerning neonatal factors, rather low correlations coefficients in Figure 1, and unclear follow-up rates. Other revisions are also needed. Including the title.
Introduction:
This provides a brief review of literature indicating impact of breastfeeding on a variety of infant and maternal outcomes and also notes a number of factors that have been found to influence initiation and continuation of breastfeeding. This study was designed to address a relative gap in research on the relationship between high-risk pregnancies and breastfeeding outcomes, as well as to analyze the association between a variety of factors, including pre-pregnancy characteristics, on the practice of exclusive breastfeeding. In the second sentence, the cited reference (#2) does not support the benefits noted, and “overall health” is not defined. In the third sentence, the reference cited (#3), mainly cites information from animal (rat) studies, and only supports the benefits for humans to a limited extent.
Response: Thank you for your observation. Regarding cited reference #2, it was replaced with one that supports the second sentence. As for the third sentence, it was modified, and a reference was cited that supports the aforementioned statement.
Materials and Methods:
The inclusion criteria indicate this study is comprised “women with high-risk pregnancies,” which includes one or more of a specific list of conditions. However, Table 2 indicates that only 55% the study population are “pre-gestational height sic (presumably high).
Response: According to the inclusion criteria, the study sample is composed of participants with conditions prior to pregnancy (pre-existing diseases) such as diabetes, hypertension, and autoimmune diseases. Participants who presented with conditions during pregnancy, childbirth, or the postpartum period were also included, as well as a group of participants with both pre-existing diseases and complications during pregnancy, childbirth, or the postpartum period. Finally, a group of participants without pre-existing diseases or complications during pregnancy was included. For the purposes of the analysis, four groups of participants were considered, and the results shown in the different tables correspond to the analysis of these four groups.
In the sample, 14.1% (80) of the participants were classified as having a normal-evolving pregnancy, while 85.9% (486) were classified as having a high-risk pregnancy, according to the definition used.
It is not clear what percent of birth population at this institution during the study timeframe was included or how representative the included population was of the broader population from which it was drawn. A consort diagram would be helpful.
Response: In accordance with your observation, the flow chart has been provided, specifying the number of participants invited, how many were included in the study, and how many completed it. In addition, the comparative statistical analysis between the group of participants and the group of non-participants is shown.
It is not clear what percent of women with one of these pre-pregnancy conditions, which were presumably identified during a third trimester visit, were actually seen at the 6-month post-partum follow-up visit, at which time information concerning the outcome variable, breastfeeding, was obtained. It would be unusual if 100% were seen at follow-up, which raises the question whether the sample was actually limited to those with follow-up, in which case, it would be important to know how those who did not follow-up compared with those who did.
Response: Thank you for your observation. The information regarding the pregnant women invited, the participants during the third trimester of pregnancy, and the percentage of mothers who completed the study is shown in the flow chart located in section 2.2, "Participants"
It is also not clear what “consecutive case sampling” refers to.
Response: This is a non-probability convenience sampling method, based on consecutive cases that met the selection criteria.
The abstract refers to “children between 6 and 48 months;” but it is not clear what data was obtained after the six-month post-partum visit or how it was used.
Response: In accordance with your observation, the phrase "children between 6 and 48 months" was removed from the abstract section, as the participants' children were only evaluated at the time of birth. Information on the introduction of complementary feeding and breastfeeding was obtained through questioning of the mother.
“Complicated” pregnancy was defined in terms of a set of conditions that developed during gestation. Mothers were classified into one of three groups, but does not appear from data in Table 2 that these were actually mutually exclusive groups, since the number of subjects in the combined “pregestational and gestational high-risk groups” is less than the sum of the component groups, suggesting that there is overlap between the pre-gestational and gestational groups and that the combined category actually reflects pre-gestational and/or gestational high risk. This needs to be clarified.
Response: In accordance with the observation, the three groups of participating mothers are not mutually exclusive. Some mothers had pre-existing conditions that justified considering the pregnancy as high-risk, and also experienced some complication during pregnancy, childbirth, or the postpartum period. The only mutually exclusive group was that of the 80 mothers who had a normal-evolving pregnancy and who presented no risk factors.
Though the birth hospital in this study is described as a “Bay and Woman Friendly” hospital, it is not clear what type of breastfeeding education and support women receive at this hospital, either prenatally during gestation, during birth admission, or on discharge.
Response: In accordance with the observation, information is provided about the type of education offered to all patients at the National Institute of Perinatology. This information is located in section 2.13, "Ethical Considerations"
“Neonatal outcomes” in 2.10 appear to include “other conditions of the baby at birth,” though not all of the conditions listed would necessarily be apparent at birth, beyond weight for age and “prematurity” (which latter is not defined or further specified). Length of stay, age at initiation of oral feeding, or neurologic conditions are not included.
Response: The newborn evaluation was performed only at the time of birth. Information on the initiation of complementary feeding was provided in the results section. Additionally, Table 2 provides information on gestational age (preterm or term delivery), the newborn's classification according to weight for gestational age, and whether or not complications were present. Information on the length of the newborn's hospital stay or their neurological conditions is not available.
No information was presented concerning the frequency/type of follow-up medical care the infants received, which could affect the ongoing support or lack thereof of breastfeeding; nor was any information provided concerning post-discharge illnesses that might have affected breastfeeding.
Response: The newborn was evaluated only at the time of birth. There is no information available on illnesses that may have arisen after hospital discharge and that could affect breastfeeding. This situation is considered a limitation of the study.
Interestingly, the outcome variable chosen for the regression model analysis was “non-exclusive breastfeeding,” which would suggest that the benefits of breast-feeding are restricted to those who are exclusively breast-fed, which is not clear from this paper or other published literature. Also, since exclusive breastfeeding is defined as exclusive breastfeeding for at least six full months, this would further imply that exclusive breastfeeding for any shorter duration does not have any benefit. Also, it is not clear whether introduction of beikost (solids) prior to six months likewise would be classified as non-exclusive breastfeeding, even if milk feeding is exclusively breast milk. The rationale for choosing non-exclusive vs. exclusive as an outcome variable needs to be explained.
Response: I completely agree with your observations. Currently, there is still controversy over whether introducing foods before the age of 6 months in mothers practicing exclusive breastfeeding means it is no longer considered exclusive, or whether non-exclusive breastfeeding is unsuitable for some mothers. In this case, for the data analysis, we adhered to the definition of exclusive breastfeeding by the World Health Organization. However, at no point did we consider that non-exclusive breastfeeding is not beneficial for the infant.
Though there are likely interactions among these variables, as briefly noted in the Discussion, no interactions are tested for in the regression analyses, though such is proposed for future research.
Response: Agreed with your observations. Due to the methodological design used, it is challenging to perform a detailed analysis of the potential interactions among the different variables. Additionally, we lack information on some other variables that could be influencing these interactions. Therefore, we propose to investigate these relationships in future studies.
Results:
Table 1 presents a chi-square comparison of exclusive breastfeeding (EBF) vs. non-exclusive across a spectrum of sociodemographic variables. Figure 1 shows Spearman’s Correlation between exclusive breastfeeding and a variety of factors that might contribute to this practice. Although 5 of 8 of these factors have statistically significant relationships, the correlation coefficients are all rather low.
Response: Thank you for your observations. We agree that the correlation coefficient values are low; however, these results indicate the relationship between these variables and exclusive breastfeeding. In future studies, we will explore how these variables may influence the duration of exclusive breastfeeding.
Table 2 shows chi-square correlations between various “medical” variables and breastfeeding practices. The categories in this table need to be left-aligned with consistent indentation of subcategories. As noted above, “pregestational height risk” needs to be changed to “pregestational high risk;” and “pregestational and gestational high risk” presumably should be “pregestational and/or gestational high risk.” “Neonate” presumably is equivalent to “baby” and only one of the two needs to be included.
Response: In accordance with your observations, the suggested changes have been made.
Table 3 presents chi-square correlations between various barriers and various facilitators and breastfeeding. As with Table 2, these items should be left aligned.
Response: In accordance with your observation, the suggested change has been made.
Table 4 shows the results of 4 progressively inclusive regression models in predicting non-exclusive breastfeeding. As with other tables, left-alignment of the individual variables for each model would be easier to read. Though many of the statistically factors have some biological plausibility as noted in the Discussion, others are less intuitive, most notably “neonatal complications,” though that factor may be due to a restrictive definition of neonatal factors, as noted above.
Response: In accordance with your observations, the suggested changes have been made.
The title of the article focuses on a comparison of pre-pregnancy conditions with obstetric complications; however, this is not discussed as such in results either in relation to data in Table 2 or Table 4, though the data in Table 4 do show that the OR for high-risk pre-pregnancy conditions is statistically significant in all four models whereas pregnancy complications is not.
Response: In accordance with your observation, a restructuring of the discussion was carried out, focusing on the comparison of pregestational conditions with obstetric complications. Additionally, information was added to Table 3 regarding high-risk pregestational conditions, high-risk pregnancy, and pregnancy without complications. Furthermore, supplementary information was incorporated into Table 2 and Figure 2 to enhance understanding.
Discussion/Conclusion:
In 4.1 reportedly 43% of women with high risk-pregnancies practiced EBF; however, it is not clear how that number ties to related data in Table 2 or in the narrative in Results.
Response: The observation is accepted. Table 2 has been corrected, and an explanation is provided regarding the different proportions of exclusive breastfeeding according to the analyzed groups (pregestational history, with pregnancy complications, those with both pregestational history and pregnancy complications, and the group of participants with normal pregnancy progression). Eighty of the participants had a normal pregnancy, of whom 46 (57.5%) reported exclusive breastfeeding.
As noted in #5 under results, some discussion concerning relative risk of pre-pregnancy conditions vs. obstetric complications is warranted. Even so, that is not the only, or even most salient, factor identified in this study. Moreover, that comparison is not included in the conclusion.
Response: In accordance with the observation, a discussion is provided regarding pregestational conditions in comparison with obstetric complications. Additionally, this comparison is included in the discussion section.
The conclusion lists some of the possible ramifications of the findings of this study, but some of that is speculative or based on other studies.
Response: In accordance with the observation, the text of the conclusions section has been corrected.

Round 2
Reviewer 1 Report
Comments and Suggestions for Authors
The authors have incorporated all of my comments. However, some technical issues still need to be addressed.
Please ensure that:
- The font is consistent in all tables and diagrams.
- The letter "n" is included everywhere (for example, it is missing next to "lack of interest" – only the number is shown).
- The formatting of "n" is consistent (decide whether to use italics or not).
- All abbreviations used in tables and figures (including flow diagram) are clearly explained, preferably in the table or figure title.
Reviewer 2 Report
Comments and Suggestions for Authors
The authors have adequately addressed most of the issues/suggestions raised by this reviewer. However, some further revision is warranted.
The addition of the Flow Diagram is particularly helpful, as well as the added narrative that describes it, though I would consider moving this under results rather than methods. However, some of the semantics in the flow diagram could benefit from rewording:
- “After delivery time”: change to “During delivery hospitalization”
- “Their address….”: “Update Medical History, Obtain Newborn information, update phone number and address for follow-up contact….”
- “At another time….”: “6 months post-partum”
- “Were locate…”: “Contact mothers and schedule appointment…“
- “Were applied…” : Complete survey regarding breastfeeding and other variables…”
- Would also suggest right justification of specific reasons for each “withdraw” box
- Loss to follow-up is actually 19.1% = 133/696 not 18.6%
In Section 2.2, the additional information regarding participants is helpful, including the new opening paragraph. However, the opening sentence should say that the
“study population was selected from women followed….” . The next paragraph should start by saying that women were classified according to a) whether they had one or more pre-pregnancy conditions that represented a real or potential risk to maternal or fetal well-being but did not develop any other complication during pregnancy; b) whether they had a complication during pregnancy; or whether they had neither. I would then move the more detailed list of pre-pregnancy conditions down to Section 2.5, which currently has a more limited list of such conditions.
Regarding Section 2.5, I think the title “high-risk pregnancy” is somewhat equivocal. On the one hand, this designation appears to be limited to “pre-pregnancy conditions that pose a risk to the health of both the mother and the fetus;” on the other hand, it also appears to include those with complications that arose during pregnancy and to those who had both “pre- and gestational conditions” (as the new title of the article suggests). The new last sentence of 2.5 indicates that “for analysis purposes, participants were classified into four groups: …” However, it is not clear whether group 1 are those women who only had pre-pregnancy conditions, group 2 only had pregnancy complications, and group 3 had either 1 or 2 or both. Moreover, group 3 in Table 2 is described as “pre-gestational and gestational high risk.” A diagram more clearly showing the overlap of groups 1-3 would help. In any case, the apparent inconsistencies need to be clarified and resolved.
In the results, the new paragraph describing breastfeeding prevalence in different groups, indicates that 312 women had pre-pregnancy risk factors, 406 had complications during pregnancy, and 486 had both. However, it would seem more likely that 486 had either pre-pregnancy or pregnancy complications or both.
Regarding Figure 1, I would suggest noting in Discussion that those correlations that are statistically significant have low coefficient values, which may explain why they are not significant in any of the regression models, except for pre-pregnancy conditions. The fact that the number of pre-pregnancy conditions also has a low correlation value (though higher than most) raises the additional question of whether pre-pregnancy conditions was included in the regression analyses as a continuous variable (as in the correlation analysis) or as a categorical variable as in Table 2. This needs further clarification/explanation.
Comments on the Quality of English LanguageSee comments to authors
Author Response
Reviewer 2
Author's Reply to the Review Report (Reviewer 2)
The authors have adequately addressed most of the issues/suggestions raised by this reviewer. However, some further revision is warranted.
We express our most sincere gratitude for all the valuable suggestions you have provided, which have significantly enriched our manuscript. We deeply appreciate the time dedicated and the generosity in sharing your expertise and knowledge, which have been fundamental in improving the quality and clarity of our work. Your contribution has been invaluable, and we are extremely grateful for your support.
The addition of the Flow Diagram is particularly helpful, as well as the added narrative that describes it, though I would consider moving this under results rather than methods. However, some of the semantics in the flow diagram could benefit from rewording:
“After delivery time”: change to “During delivery hospitalization”
“Their address….”: “Update Medical History, Obtain Newborn information, update phone number and address for follow-up contact….”
“At another time….”: “6 months post-partum”
“Were locate…”: “Contact mothers and schedule appointment… “
“Were applied…”: Complete survey regarding breastfeeding and other variables…”
Would also suggest right justification of specific reasons for each “withdraw” box
Loss to follow-up is actually 19.1% = 133/696 not 18.6%
Response: The observation is accepted. The flowchart has been moved to the results section, and the terms used within it have also been reformulated.
The percentage of loss to follow-up was verified, confirming that it is 18.67%, which corresponds to 130 losses out of a total of 696 participants invited to participate.
In Section 2.2, the additional information regarding participants is helpful, including the new opening paragraph. However, the opening sentence should say that the “Study population was selected from women followed….”. The next paragraph should start by saying that women were classified according to a) whether they had one or more pre-pregnancy conditions that represented a real or potential risk to maternal or fetal well-being but did not develop any other complication during pregnancy; b) whether they had a complication during pregnancy; or whether they had neither. I would then move the more detailed list of pre-pregnancy conditions down to Section 2.5, which currently has a more limited list of such conditions.
Response: Thank you for the observation. We have modified the first paragraph according to the reviewer's suggestion. In the following paragraph, it is indicated that the women were classified according to…
The detailed list of pre-pregnancy conditions and some of the most frequent complications that occurred during pregnancy, childbirth, or postpartum has been moved from section 2.2 to section 2.5.
Regarding Section 2.5, I think the title “high-risk pregnancy” is somewhat equivocal. On the one hand, this designation appears to be limited to “pre-pregnancy conditions that pose a risk to the health of both the mother and the fetus;” on the other hand, it also appears to include those with complications that arose during pregnancy and to those who had both “pre- and gestational conditions” (as the new title of the article suggests). The new last sentence of 2.5 indicates that “for analysis purposes, participants were classified into four groups: …” However, it is not clear whether group 1 are those women who only had pre-pregnancy conditions, group 2 only had pregnancy complications, and group 3 had either 1 or 2 or both. Moreover, group 3 in Table 2 is described as “pre-gestational and gestational high risk.” A diagram more clearly showing the overlap of groups 1-3 would help. In any case, the apparent inconsistencies need to be clarified and resolved.
Response: The observation is accepted. The title of section 2.5 has been modified according to the reviewer's suggestion.
The last sentence of section 2.5 has been moved to section 2.2, where a diagram is included that clearly shows the overlap of groups 1, 2, and 3.
In the results, the new paragraph describing breastfeeding prevalence in different groups, indicates that 312 women had pre-pregnancy risk factors, 406 had complications during pregnancy, and 486 had both. However, it would seem more likely that 486 had either pre-pregnancy or pregnancy complications or both.
Response: The suggestion is accepted. The prevalence of exclusive breastfeeding has been clarified according to the group to which the participants belonged.
Regarding Figure 1, I would suggest noting in Discussion that those correlations that are statistically significant have low coefficient values, which may explain why they are not significant in any of the regression models, except for pre-pregnancy conditions. The fact that the number of pre-pregnancy conditions also has a low correlation value (though higher than most) raises the additional question of whether pre-pregnancy conditions was included in the regression analyses as a continuous variable (as in the correlation analysis) or as a categorical variable as in Table 2. This needs further clarification/explanation.
Response: The suggestion is accepted. The results of the different regression models are discussed in relation to Figure 1 and Table 2 as follows:
In our study, the bivariate correlation analysis with continuous variables (Figure 1) revealed generally low correlations between the predictors of breastfeeding practices, with the exception of pre-pregnancy risk. This finding could explain the lack of statistical significance of the regression models with breastfeeding as the dependent variable. Similarly, the analysis of the same categorized variables (using dummy variables) yielded consistent results. These findings suggest that the absence of association is not attributed to the coding of the variables, which reinforces the robustness of our results.
